# Current and Potential Cognitive Development in Healthy Children: A New Approach to Raven Coloured Progressive Matrices

**DOI:** 10.3390/children9040446

**Published:** 2022-03-22

**Authors:** Pietro Smirni, Daniela Smirni

**Affiliations:** 1Department of Educational Sciences, University of Catania, 95124 Catania, Italy; pietrosmirni@hotmail.it; 2Department of Psychology, Educational Science and Human Movement, University of Palermo, 90128 Palermo, Italy

**Keywords:** zone of proximal development, problem-solving abilities, cognitive development, fluid intelligence, intellective test, neuropsychological assessment, analogical reasoning

## Abstract

In clinical practice and research, Raven’s Coloured Progressive Matrices (RCPMs) continue to be used according to a single procedure that aims to evaluate a single overall score of the current general intelligence level. This study aimed to examine potential cognitive development in a sample of 450 typically developing children, aged from 6 to 10 years, by administering RCPMs according to the standard procedure followed immediately by a standardized interview on incorrect items. In addition, the study aimed to analyze how performance differed across age groups. The results analysis was examined on the basis of three different factors in which the items were grouped in previous factorial studies. The results found that performance improved markedly and significantly after the interview; however, the improvement was not homogeneous in the three factors across age groups or within each age group. The age groups showed a different development potential in relation to the nature of the task: the younger ones showed a greater increase on items requiring figure completion, and the older ones showed a greater increase on analogical reasoning items. Finally, the children who showed the greatest improvement were those with the best performance in standard RCPM administration. The procedure described in the present research could represent a useful tool in clinical practice and in the research for a broader cognitive assessment focused on potential cognitive development, as well as on real cognitive development, and to favor the planning of more adequate rehabilitation and educational treatments.

## 1. Introduction

Raven’s Progressive Matrices were developed in the 1930s as a ‘culture free’ and nonverbal test to study the genetic and environmental determinants of ‘general intelligence’ (the g factor) [1]. Afterward, to provide a simplified format for children and the elderly, the Raven’s Coloured Progressive Matrices (RCPMs) were devised [2,3]. Both tests were included in the theoretical framework of Spearman factorial analysis studies [4,5], according to which all intellective tests correlate positively, albeit to a different extent.

Indeed, both the tests were designed to measure such a hypothetical construct of ‘general intelligence’ underlying all intellective performances as the ability to identify logical relationships within different elements and to generate the abstract organizing rules [1].

Over time, the Raven tests have been widely used as one of the best psychometric measures of general intelligence in the clinical, educational, and professional settings, providing a domain-independent measure of fluid reasoning by a single total score [6,7]. However, Raven himself pointed out that the various sets of items may have different cognitive features such as simple or discrete pattern completion, pattern completion with closure, and concrete or abstract reasoning by analogy. Moreover, several factorial analysis, lesion, and neuroimaging studies highlighted the involvement of different brain structures and different cognitive abilities [8,9,10], such as abstract/analytic reasoning and visuoperceptual or figural reasoning [11,12]. Neuroimaging studies reported that items involving analytic reasoning tend to activate bilateral frontal areas and left temporal, parietal, and occipital regions, while items involving visuospatial reasoning activate right frontal and bilateral parietal brain areas [13,14,15,16,17,18,19,20]. In addition, there is evidence for different consequences of anterior and posterior lesions [13,17,18,21].

However, in daily clinical practice, RCPMs continue to be used with a single total score, where ‘zero’ is the wrong answer and ‘one’ is the right answer, referring to the current level of general intelligence. Opposed to such psychometric-based assessment, Vygotsky devised the construct of a ‘zone of proximal development’ (ZPD) to point out the distance between the ability to solve problems independently and the ability to solve problems with support. The ZPD construct would include both the reached and attainable cognitive and developmental level, as potential for emerging ‘*the area of immature, but maturing processes*’ [22]. Moreover, according to Vygotsky, the size of the ZPD was more predictive than intelligence quotient (IQ) and allowed one to ‘*gain the potential for directly studying that which most precisely determines the level of mental maturation that must be completed in the proximal or subsequent period of his age development*’ [23,24]. 

Therefore, from a Vygotskian perspective, neuropsychological assessment should focus on two different cognitive levels: the current and the potential one. According to Vygotsky’s ZPD theory, children may not solve a problem on their own, but they could do it with external support. Additionally, individual children may achieve similar scores on the standard testing, but they may have different results when support is provided [22]. Several authors have developed ‘dynamic’ diagnostic methods, such as the learning test [25,26] and the graduated prompt procedures [27,28,29,30,31], where standard assessment is enhanced by more supportive and interacting examiners using exploratory talk, cues, and suggestions and providing feedback or training instructions to enable the performance on problem-solving tasks, and to evaluate the learning potential as the ‘ability to learn’ after cues. Binet himself [32] already pointed out that the intelligence test he devised did not always provide an adequate picture of the child’s cognitive potential development which, on the other hand, could be achieved through items showing how to learn [33]. 

A dynamic assessment, therefore, provides more complete information on the cognitive and learning abilities and potential level of intellectual functioning than the standardized and ‘static’ IQ testing, where instructions are given without any support, according to the manual, and the children either answer correctly or are wrong, and what a child potentially can achieve is neglected or underestimated. Therefore, an accurate profile of intellectual functioning should include both dynamic and standard measures [27,34,35].

In the current study, in line with previous studies [8,9,10], the RCPM scores were examined according to three different qualitative item clusters factorially extracted: (1) Factor III involves the perceptual analysis of the elements as a continuous pattern [8,9,10,11,19,36,37]. Detecting the missing element results from an association of similarities between the target figure and the entries, and identical features are mapped onto one another (e.g., B1 or B2). It is very likely, therefore, that such similarity tasks may be mainly supported by basic visuospatial attentive abilities [38] and errors can be due to a holistic visual–perceptual approach focusing on a prominent perceptual feature. (2) Factor I implies continuous and discrete pattern completion through closure. The choice of the missing element to form a coherent gestalt is a complex perceptual, analytical process, which requires a gestalt reasoning; a simultaneous visuoperceptual analysis of the target and the response alternatives, as a whole and as individual components; and a flexible shifting part–whole and whole–part [8,9,10,19,39,40]. (3) In Factor II, involving abstract reasoning by analogy [19,39,41,42], problem solving results from an abstract process of analogical reasoning, implying the ability to understand the logical relational similarity between the three given elements and the entries, according to the paradigm ‘a is to b as c is to (entry 1, 2, 3, 4, 5, or 6)’. The wrong answer results from an associative choice focused on a concrete association between only one of the three given elements and one of the six entries. 

The current study aimed to provide an overview of the performance in a sample of healthy children on the RCPMs by means of descriptive statistics, administering the test according to the standard procedure followed immediately by a resubmission of the wrong items along with a structured interview.

## 2. Materials and Methods

### 2.1. Participants

A sample of 450 healthy children participated in the study, 222 girls and 228 boys, subdivided into 5 age groups ranging from 6 to 10 years. Recruitment was continued until a balanced number of subjects was reached and until a balanced female/male distribution was reached for each age group. 

Participants were recruited from the children population in 13 Sicilian municipalities; they were of medium social status, with no history of neurological or psychiatric diagnosis, learning disability, or developmental delay. 

This study is part of a larger previous study aimed at evaluating the neurodevelopment of cognitive functions in healthy children, which had been approved by the local ethics committee and school board [8]. Only children whose parents voluntarily provided informed consent to participate were recruited. The study was approved by the Bioethics Committee of the University of Palermo. The research was conducted following the ethical principles of the Declaration of Helsinki.

Children or parents who did not agree to take part in the study were excluded.

### 2.2. Procedures

The RCPMs were administered by expert clinical neuropsychologists individually, in paper format, without time limit, as indicated by Raven [3]. The standard test requires that in a visual pattern problem a missing element be chosen among six possible choices. One score was given for each correct answer. Immediately after the standard administration, a standardized interview was given only for the wrong items. After the standardized interview, an additional score was recorded including the sum of the previous standard score and the number of errors corrected. From now on, we will call the first the standard score condition and the second the ZPD score condition.

The results were examined according to three qualitative clusters of items previously identified referred to the cognitive processes involved [8,9,10]: Factor I, called ‘pattern completion through identity and closure’, composed of 15 items: A7, A8, A9, A10, Ab4, Ab5, Ab6, Ab7, Ab8, Ab9, Ab10, Ab11, B3, B4, and B5; Factor II, called ‘closure and abstract reasoning’, composed of 10 items: A11, A12, Ab12, B6, B7, B8, B9, B10, B11, and B12; and Factor III, called ‘simple pattern completion’, composed of 11 items: A1, A2, A3, A4, A5, A6, Ab1, Ab2, Ab3, B1, and B2.

Figure 1 shows a graphic description of the standardized interview including the following questions in Factor III and Factor I: (1) ‘Do you remember what answer you gave me a while ago in this task?’ When the subject reconfirmed the previous wrong answer, he or she was told: (2) ‘Do you still think this is the correct answer?’ When the subject did not reconfirm the previous wrong answer and gave the correct answer, he or she was told: (3) ‘Before you told me the number…. Now you tell me the number…. Which of the two answers seems to you the correct one?’ When the subject gave another incorrect answer, he or she was told: (4) ‘Before, you told me the number…. Now you tell me the number…. Which of the two answers seems to you the correct one?’ (Figure 1a).

As the analogical reasoning tasks of Factor II require the identification of a reasoning criterion, the following slightly different standardized questions were given for the wrong Factor II items: (1) ‘Do you remember what answer you gave me a while ago in this task?’ When the subject reconfirmed the previous wrong answer, he or she was told: (2) ‘What reasoning did you do?’ When the subject’s criterion was wrong, he or she was told: (3) ‘Do you think there might be another criterion?’ When the subject did not reconfirm the previous wrong answer and gave the correct answer, he or she was told: (4) ‘Before you told me the number…. Now you tell me the number…. Which of the two answers seems to you the correct one?’ (Figure 1b).

In no case did the interview questions explain the underlying rules, nor did they suggest answers, nor did they provide direct support for solving the problem.

### 2.3. Statistical Analysis 

In each age group, two different mean values of the RCPM performance were calculated in the three qualitative clusters of items: the values of the standard score condition and the values of the ZPD score condition, as the sum of the wrong items corrected after the interview and the correct items of the standard procedure.

In Factors I and II, both in the standard condition and in the ZPD condition, modality (value) multiplied by frequencies (number of observations) was calculated and summed (Σxi*f) in each age group. The differences between the sum modalities multiplied by frequencies in the standard and in the ZPD conditions were calculated, and the amount of increase was obtained in absolute (∆ values) and percentage value (∆% values). In addition, a paired *t*-test was performed to provide further indication of the differences between means of performance in the standard condition and in the ZPD condition.

Furthermore, in order to verify whether and to what extent the improvement after the interview was related to the current cognitive development, each age group was divided into two subgroups: below and above mean performers in the standard condition in Factors I and II. Absolute and percentage global increases after the interview were calculated in subgroups below and above the mean in the standard administration of RCPMs. In the same subgroup of subjects below/above mean, the Pearson correlation coefficient was calculated between RPCM scores in the standard condition and in the ZPD condition. 

## 3. Results

Table 1 shows the mean values in the RCPMs of each age group in the three factors in the standard condition and after the structured interview (ZPD condition). 

As expected, in each age group and in the three factors, the number of correct items gradually increased in both conditions with increasing age. Moreover, after the interview, performance improved across the groups and the three factors, but to a different extent. More specifically, in Factor II, the improvement was lower than the other two factors and no group reached the maximum scores. In Factor III, from the age of eight, there were no wrong responses and each group reached a maximum improvement, compared to the standard condition. In the two youngest groups, the number of errors adjusted was close to the maximum, while in the oldest groups all errors were corrected. Therefore, the Factor III performance was very high for the whole sample and poorly discriminating at different ages, and it was not considered in subsequent analyses.

Table 2 and Table 3 show the frequency distribution and the sum modalities multiplied by frequencies (Σxi*f) in the standard condition and in the ZPD condition of each age group in Factor I and Factor II. In the standard condition, in both factors, the frequencies were distributed over a wide range. In the youngest group, more than 80% were distributed around lower values, while in the oldest group, more than 95% were distributed around higher values. Conversely, in the ZPD condition, in Factor I the frequencies were shifted around the higher values in all groups, while in Factor II the frequencies continued to be distributed over low values in the youngest group and on higher values in the oldest group.

Table 4 shows the increase values in the two factors in each age group. The increase was the difference, in absolute and percentage values, between the sum of modalities multiplied by frequencies in standard and ZPD condition (∆ and ∆% values).

In Factor I, the younger groups, 6–7 years, showed a greater increase than the older ones. The percentage improvement gradually decreased from 24.5% at 6 years to 12.6% at 10 years. Conversely, in Factor II, the younger groups showed very low improvements after the interview with an increase of about 6.7% at 6 years and 13.2% at 7 years, and they were unable to resolve abstract analogical items even after the interview. Instead, the older groups, after the interview, showed a greater improvement with a percentage increase of 17.0% at 8 years, 25.6% at 9 years, and 37.5% at 10 years. 

Table 5 shows the results of the *t*-tests on performance in RCPMs between the two conditions. The differences were statistically significant in each age group, with better performance in the ZPD condition. In Factor I, *t*-test values were higher in younger ages, while in Factor II, *t*-test values were higher in older groups. 

Table 6 shows the increase after the interview in each subgroup below/above the mean performers and total age-groups including both below/above the mean performers in the standard administration of RCPMs. Performance improved in both subgroups and in both factors. However, in Factor I, the below mean subgroup mostly used the suggestions of the interview, reaching an increase of more than 50%. In Factor II, on the other hand, the above mean subgroup used the provocations of the interview to a greater extent, with an increase ranging from 54.5% for the youngest to 80% at 10 years, while the subgroup below mean increased to a more modest extent among 45% and 19.1%.

Table 7 shows the correlations between the performance in the standard condition and after the interview, in each subgroup below/above the mean performers and total age-groups including both below/above the mean performers. The correlations were all positive. However, the most solid and statistically significant were those of children who performed better in the standard condition.

## 4. Discussion

The current study aimed to provide an overview of the performance on the RCPM by means of descriptive statistics, administering the test according to the standard procedure followed immediately by a resubmission of the wrong items along with a structured interview. In line with a large body of literature, the study assumed that performance after some form of support (hints, feedback, prompts), compared to the spontaneous results in a standard administration, can provide wider qualitative evidence on the child’s potential intellectual development and the cognitive modifiability [22,30,43,44,45,46,47,48,49,50,51,52,53,54].

The RCPM results were examined according to the qualitative factor subdivision that previous studies have proposed [8,9,10]. As expected, in each factor and age group, performance significantly improved after the interview. More specifically, in Factor III, implying simple or discrete pattern completion by a comparison process [19,39,41], the ceiling effect in ZPD condition was reached by the age of eight, while the youngest groups improved significantly with values close to the maximum. 

In line with the Vygotskian construct of the development potential zone [22,41,55], therefore, the interview on incorrect answers acts as a drive to recover inattentive errors, allowing the identification of errors related to an inattentive approach and errors due to a real lower cognitive ability and providing a more proper and complete cognitive profile.

Factor I proved to be more demanding than Factor III. In each age group, performance in the standard condition was poorer than that in Factor III, and no age group reached the maximum. Likewise, after the interview, improvement was lower and did not reach the maximum level in any group. However, the performance improved in each age group, but to a different extent. The higher increase was among the younger groups. It is likely that, in the standard condition, the youngest processed stimuli according to a holistic approach and were less likely to analyze stimuli into their components, in line with a long tradition of theory of reasoning proving that younger children, compared to older ones, tend to be holistic in object perception and oriented to context as a whole [56,57,58,59,60,61,62,63]. The interview might support the overcoming of the global holistic approach and activate a more accurate analytical thinking and, as a consequence, a performance improvement, at least in subjects with a cognitive potential development higher than the current. The oldest groups, on the other hand, improved to a smaller extent, probably because their performance was already higher in the standard condition and, therefore, the chances of improvement after the interview were reduced. 

In the examined sample, the performance in Factor II was lower than that in the other two factors, in line with studies proving that in the early stages of middle childhood, before age eight, in abstract reasoning processes there are still attributional interpretations rather than relational interpretations [64]. However, even in this factor, in each group, the performance improved in the ZPD condition, although to a lower extent than in the other two factors. The interview did not lead to significant improvements in the two younger groups, under the age of eight. They were unable to solve analogical reasoning problems, either independently or with the support of the interview, probably because the Factor II stimuli were even more demanding than their current and potential cognitive development. Conversely, the older children were not able to independently solve abstract analogical problems; however, they increased their performance after the interview. The improvement becomes higher around the age of 8–10. Such age-related increase was consistent with studies showing middle childhood as a crucial age span for the development of concrete operational thought, the conservation and reversibility process, and the ability to overcome the concrete perceptual datum and establish analogical relations, with a relevant increase at about 8 years [62,63,64,65,66,67,68,69].

Analogy studies suggest that the key core of analogical reasoning is the ability to understand a higher-order relational structure including multiple and different elements based on their role in the structure, beyond surface appearance. The systematic increase in analogical reasoning throughout middle childhood results from an increasing shift from mainly perceptual information processing to relational information processing [62,70,71] and a growing ability to control irrelevant perceptual distractors [72]. 

Moreover, studies on brain development proved that at around eight years of age, significant structural and functional changes [73,74] affect the whole brain and the gray matter volume [75,76,77], synaptic pruning processes [78], and functional connectivity [79,80]. Furthermore, studies document, at around age ten, a significant increase in cortical thickness in parietal and frontal association areas, as well as in higher-order cortical areas such as the dorsolateral prefrontal cortex and the cingulate cortex [75,81,82]. Therefore, such brain developmental changes are likely to support greater cognitive potential development in the 8–10-year-old groups than in the younger ones, which results in a greater increase in performance after the interview.

It is interesting to point out that when dividing each group in relation to performance in the RCPM standard procedure (below or above the mean), performance after the interview improved in both below and above mean subjects in each age group, in both factors. However, the increase was higher in subjects with a higher level of current cognitive development in standard RCPM administration. They were, therefore, the best who made the most of the suggestions of the interview. It would appear that a higher current cognitive development supports a higher cognitive potential development in healthy children. 

These results are consistent with a large body of literature [55] assuming that children who improve after prompting have better problem-solving skills than expressed by their score in the standard assessment, while children who do not improve after prompting are very likely to have a cognitive development consistent with the scores in the standard procedure. 

The first group, after promptings, consciously identify the correct choice, appearing to be able to use the experience of suggestion cueing and to be close to changing their own cognitive structures in a more advanced cognitive development. Therefore, such groups may be involved in more demanding rehabilitative or educational programs than the conventional procedure would suggest, and if properly stimulated, they can reach more advanced levels than those exhibited in a simple standard evaluation [40].

The second group, instead, does not capture suggestions and probably continues to be strongly conditioned by perceptually pregnant details and concrete thinking, which do not allow understanding the closing rules of a gestalt (Factor I) or the logical–abstract relationships between elements (Factor II). According to the literature [40,83,84,85], therefore, it is more correct for this group to be involved in a less demanding educational or rehabilitative training to avoid failures and frustrating experiences for their self-esteem. 

Therefore, the focus on potential cognitive development enhances the standard scoring system, providing a clinically more effective cognitive profile and suggestions in the clinical and rehabilitative contexts. 

Nevertheless, in both clinical practice and research, RCPMs continue to be used according to a single procedure and rated as a single overall score of general intelligence [49]. To overcome such limitation, in the current study, immediately after conventional administration, a new alternative assessment procedure was added. According to such procedure, neuropsychological cognitive assessment does not result in a simple global achievement score, but it has been expanded in talking about the errors, their corrections or persistence, the ways of recovery of incorrect answers, and the metacognitive awareness and verbalization of the thinking processes and cognitive strategies [86].

Several limitations of the current study should be considered. The study focused only on a large sample of typically developing children. Future research is needed to provide in clinical populations the effectiveness of the qualitative procedure described, in order to identify the rehabilitation potential.

Moreover, the study did not show a neuropsychological profile of the group that does not improve after the standardized interview. Further research is needed to identify the individual cognitive features of these subjects. Similarly, it would be useful to have an instrument that allows the investigation of concurrent validity.

## 5. Conclusions and Practical Implications

The presented approach to RCPMs, therefore, enhanced standard psychometric administration and scoring with a dynamic testing condition focused on the zone of proximal development, in line with Vygotsky’s theory [22], and provided both the current cognitive development and the child’s modifiability following additional prompts or suggestion cueing. Moreover, it allows elaborating tailored educational or cognitive rehabilitative programs to maximize the developmental potential for each child. Such qualitative procedure consists of the following steps:Standard administration of RCPMs, as Raven’s procedure.Standard scoring, as zero or one for wrong or correct answer.Structured interview, following standard administration, aimed at questioning each incorrect item in the standard condition.Recording of the correct answer, when, in the interview, the wrong answer is disconfirmed and the correct response is given.Distribution of both the results according to the three qualitative clusters of items.

According to the reported procedure, two different scores will be available in each of the three factors: the score in the standard procedure (as actual cognitive development index) and the score after the interview (as proximal cognitive development index). This procedure could represent a useful tool in clinical practice and in the research for a broader assessment focused on potential cognitive development, as well as on real cognitive development, and to favor the planning of more adequate rehabilitation and educational treatment programs.

## Figures and Tables

**Figure 1 children-09-00446-f001:**
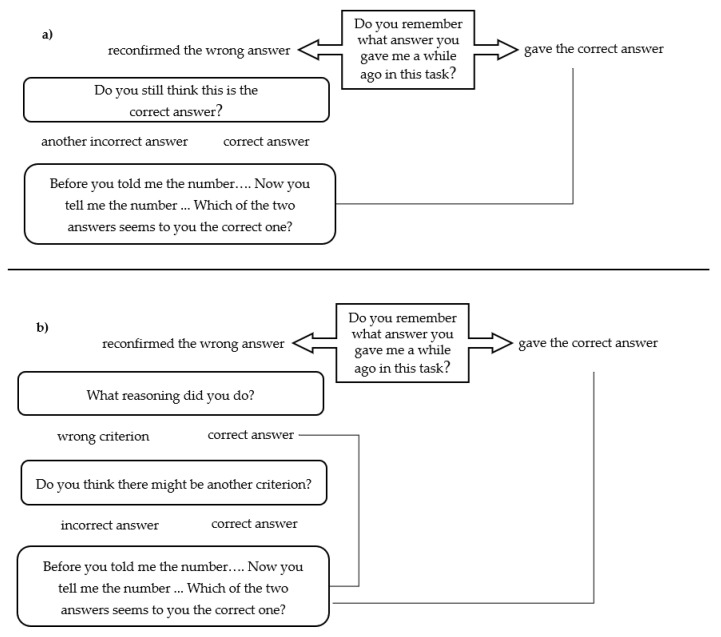
Flowchart of the standardized interview: (**a**) Factor III and Factor I and (**b**) Factor II.

**Table 1 children-09-00446-t001:** Mean values of the RCPMs in the three factors in the standard and in the ZPD conditions.

			Factor III (11 Items)	Factor I (15 Items)	Factor II (10 Items)
Age	n (f/m)	Condition	Mean ± SD	Mean ± SD	Mean ± SD
6	88 (46/42)	standard	10.44 ± 0.56	8.14 ± 1.13	1.90 ± 1.10
		ZPD	10.89 ± 0.30	10.43 ± 1.30	2.63 ± 1.52
7	90 (43/47)	standard	10.64 ± 0.60	9.20 ± 1.37	2.44 ± 1.24
		ZPD	10.91 ± 0.44	11.32 ± 1.23	3.84 ± 1.59
8	91 (46/45)	standard	10.83 ± 0.37	10.74 ± 1.42	2.93 ± 1.34
		ZPD	11.00 ± 0.00	12.56 ± 1.26	4.73 ± 1.11
9	87 (41/46)	standard	10.89 ± 0.30	12.29 ± 1.36	4.07 ± 0.91
		ZPD	11.00 ± 0.00	14.14 ± 1.13	6.89 ± 0.96
10	94 (46/48)	standard	10.92 ± 0.26	13.24 ± 1.05	4.80 ± 0.87
		ZPD	11.00 ± 0.00	14.35 ± 0.84	8.62 ± 1.19
Overall	450 (222/228)	standard	10.75 ± 0.48	10.74 ± 2.28	3.24 ± 1.53
		ZPD	10.97 ± 0.25	12.57 ± 1.92	5.35 ± 2.52

Legend: ZPD: zone of proximal development; n: number; SD: standard deviation.

**Table 2 children-09-00446-t002:** Factor I: frequency distribution of the correct items and the sum modalities multiplied by frequencies in the standard and ZPD conditions in age groups.

Age/Frequency		6	7	8	9	10	11	12	13	14	15	n	Σxi*f
Standard	6	2	25	33	20	4	3	1	0	0	0	88	716
	7	0	3	36	16	16	13	6	0	0	0	90	828
	8	0	0	9	11	12	28	26	4	1	0	91	977
	9	0	0	1	2	7	12	18	35	10	2	87	1069
	10	0	0	0	0	3	4	8	36	38	5	94	1245
ZPD	6	1	2	2	7	38	23	10	4	1	0	88	918
	7	0	0	1	4	19	24	28	11	3	0	90	1019
	8	0	0	0	1	4	8	34	26	10	8	91	1143
	9	0	0	0	0	1	2	5	13	21	45	87	1230
	10	0	0	0	0	0	1	3	7	34	49	94	1349

Legend: ZPD: zone of proximal development; n: number; Σxi*f: sum modalities multiplied by frequencies.

**Table 3 children-09-00446-t003:** Factor II: frequency distribution of the correct items, mean values, and the sum modalities multiplied by frequencies in the standard and ZPD conditions in age groups.

Age/Frequency		0	1	2	3	4	5	6	7	8	9	10	n	Σxi*f
Standard	6	3	38	21	19	5	2	0	0	0	0	0	88	167
	7	2	16	38	16	11	6	1	0	0	0	0	90	220
	8	0	16	15	35	13	9	2	1	0	0	0	91	267
	9	0	0	2	18	46	16	3	2	0	0	0	87	354
	10	0	0	0	6	25	49	10	4	0	0	0	94	451
ZPD	6	3	25	12	25	11	9	3	0	0	0	0	88	231
	7	0	8	10	22	15	21	11	3	0	0	0	90	346
	8	0	1	1	9	22	41	13	3	1	0	0	91	430
	9	0	0	0	0	0	7	18	45	13	3	1	87	590
	10	0	0	0	2	0	1	4	2	9	71	5	94	811

Legend: ZPD: zone of proximal development; n: number; Σxi*f: sum modalities multiplied by frequencies.

**Table 4 children-09-00446-t004:** Increase in Factor I and Factor II: differences between the sum modalities multiplied by frequencies in the standard and in the ZPD conditions in absolute and percentage values.

		Factor I				Factor II			
Age	n	Σxi*f Standard	Σxi*f ZPD	∆	∆%	Σxi*f Standard	Σxi*f ZPD	∆	∆%
6	88	716	918	202	24.5	167	231	64	6.7
7	90	828	1019	191	23.2	220	346	126	13.2
8	91	977	1143	166	20.1	267	430	163	17.0
9	87	1069	1230	161	19.5	354	590	236	25.6
10	94	1245	1349	104	12.6	451	811	360	37.5
Total	450	4835	5659	824	100	1459	2048	949	100

Legend: ZPD: zone of proximal development; n: number; Σxi*f: sum modalities multiplied by frequencies; ∆: difference between standard and ZPD Σxi*f; **∆**%: difference value compared to total difference * 100.

**Table 5 children-09-00446-t005:** Factor I and Factor II: *t*-test comparing differences between RPCM mean scores in the standard condition and in the ZPD condition.

		Factor I			Factor II		
Age	n	*t*-Test	d.f.	*p*-Value	*t*-Test	d.f.	*p*-Value
6	88	12.40	174	<0.001	3.63	174	<0.001
7	90	10.88	178	<0.001	6.57	178	<0.001
8	91	9.09	180	<0.001	9.81	180	<0.001
9	87	9.70	172	<0.001	16.77	172	<0.001
10	94	7.96	186	<0.001	25.76	186	<0.001

Legend: d.f.: degrees of freedom; n: number.

**Table 6 children-09-00446-t006:** Factor I and Factor II: increase after interview in subgroups below/above the mean performers and in total age-groups.

	Age	n Below	n Above	Increase Below	Increase Above	Increase Total	Increase % Below	Increase % Above
Factor I	6	50	38	114.8	87.2	202	56.8	43.2
	7	48	42	101.9	89.1	191	53.3	46.7
	8	47	44	85.7	80.3	166	51.6	48.3
	9	42	45	77.7	83.3	161	48.3	51.7
	10	45	49	49.8	54.2	104	47.9	52.1
Factor II	6	40	48	29.1	34.9	64	45.4	54.5
	7	40	50	56.0	70.0	126	44.4	55.5
	8	33	58	59.1	103.9	163	36.2	63.7
	9	25	62	70.4	174.6	245	28.7	71.2
	10	18	76	68.7	290.3	359	19.1	80.8

Legend: n: number.

**Table 7 children-09-00446-t007:** Factor I and Factor II: correlation between RPCM scores in the standard condition and in the ZPD condition in subgroups below/above and in total age-groups.

	Age	n Below	n Above	r Below	r Above	r Total
Factor I	6	50	38	0.16	0.65 ***	0.42 ***
	7	48	42	0.37 **	0.61 ***	0.55 ***
	8	47	44	0.46 ***	0.55 ***	0.56 ***
	9	42	45	0.32 *	0.54 ***	0.62 ***
	10	45	49	0.22	0.45 **	0.27 ***
Factor II	6	40	48	0.44 **	0.68 ***	0.82 ***
	7	40	50	0.41 **	0.68 ***	0.67 ***
	8	33	58	0.58 ***	0.65 ***	0.63 ***
	9	25	62	0.55 ***	0.63 ***	0.62 ***
	10	18	76	0.45	0.63 ***	0.49 ***

Legend: n: number; r: correlation coefficient; *p*-value: * < 0.05, ** < 0.01, *** < 0.001.

## Data Availability

The data presented in this study are available on request from the corresponding author.

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
