# Peer review of "Current and Potential Cognitive Development in Healthy Children: A New Approach to Raven Coloured Progressive Matrices"

_children, 2022, doi:10.3390/children9040446_

Round 1

Reviewer 1 Report

+ Central idea is quite interesting. 

+ Abstract should be clearer about objectives, findings and implications.

+ The fact of not using a concurrent validity instrument should be presented as a major limitation of the study. 

+ The references should be accurately updated btween the period 2019-2022.

Ortega, V. H. (2021). Validez del test de matrices progresivas Escala coloreada de Raven en estudiantes de 6 a 11 años. Warisata-Revista de Educación3(7), 10-27.

Author Response

+ Central idea is quite interesting. 

Authors’ reply: we thank the reviewer for this positive comment

+ Abstract should be clearer about objectives, findings and implications.

Authors’ reply: thanks for this suggestion, we have modified accordingly (lines 10-26).

+ The fact of not using a concurrent validity instrument should be presented as a major limitation of the study. 

Authors’ reply: We agree with the reviewer, if a concurrent validity instrument were possible the study would be much more robust. We have added as a limitation as suggested (line 806).

+ The references should be accurately updated between the period 2019-2022.

Ortega, V. H. (2021). Validez del test de matrices progresivas Escala coloreada de Raven en estudiantes de 6 a 11 años. Warisata-Revista de Educación3(7), 10-27.

Authors’ reply: thank you for this useful suggestion which helps us to update the literature on RCPM even in the most recent period (between 2019 and 2022). We added reference n.7.

Reviewer 2 Report

This work evaluates a potential cognitive index of the RCPM in a sample of healthy children. The findings describe different improvement trajectories according to the test's specific Factors and age range and indicate that children with higher standard scores are the ones benefiting the most from prompting, in line with previous evidence.

The manuscript is clear in most of its parts.

However, the way in which some claims were made throughout the manuscript is not appropriate given that this work followed a qualitative approach. I’ll elaborate this issue further in the comments.

Introduction

Line 30: the expression “discrimination at lower levels” is not clear to me. Can the authors rephrase this sentence?

Line 51: this sentence is missing the related references.

Line 70: the authors hypothesized that the interview would significantly increase the performance. However, such hypotheses cannot be tested with the approach used here, but applying statistical analyses. It would be more appropriate to claim that the aim of the study was to provide an overview of the performance on the RCPM by means of descriptive statistics.

Moreover, I suggest adding a part on dynamic testing.

Materials and Methods

Participants

Line 81: How can the number of subjects be counterbalanced? Does it mean that the authors kept recruiting until a certain number per age-group was reached?

Procedures

Line 97: the terms Factor III and I appear here without an introduction or any explanation on what they are. I suggest adding in the method a brief description of the test, the factors and to which cognitive abilities and strategies they have been associated with in previous research.

Line 97-115: I have two comments regarding the interview. First, I think it may be helpful to provide a graphic description of the procedure, for example using a flow chart, which I believe would make it quicker and easier to understand how it was conducted. But this a matter of personal preference, so if the authors prefer a verbal description, I understand their choice. Second, provide a rationale of why the structure of the interview for Factor II was different to the one of the interview for Factor I and III, for which children were not asked to explain the reason behind their response, but they were only given time to rethink their initial choice. Do the authors think that this difference may have influenced the results? If so, how?

Line 121-126:  Please add the maximum score for each Factor to simplify the understanding of Table 1.

Results

Table 1: please reorder the Factors in the table.

Line 151: The claims made here are in line with the use of descriptive statistics. However, a simple t-test – for example – would go beyond the “mere” description of higher score in the ZPD as compared to the standard condition, providing a more solid indication of whether an actual improvement has occurred or not.

Discussion

Line 212-213: “performance significantly improved after interview” this claim can’t be made based on descriptive statistics. The same goes for similar claims on differences among age-groups.

Line 227-230: this part looks more of a conclusion, in my opinion, thus I suggest moving it towards the end of the discussion.

Line 273-276: this sentence repeats what has already been said above.

Line 306-308: in line with my comment on the t-test, here a simple correlation would verify if, indeed, subjects with the highest improvement are the ones with the highest standard score, although Table 5 provides a similar indication.

Line 325: Do the authors mean “less intensive” or “less demanding” with the term lower?

Conclusion

Line 342: the authors refer to prompts, suggestions and cuing. I think it would be helpful to add a part, maybe in the introduction, on these procedures, on how they are used in neuropsychological testing and how they may differently influence the ZPD.  

Line 356: I’d add as a limitation the fact that only descriptive statistics were computed.

Overall comment

This work has the potential to be of interest not only for the clinical audience but also for researchers in the field of cognitive sciences. However, the meaningfulness of the findings is strongly limited by how the data were analyzed. Sometimes researchers try to make inferences from data or analyses that cannot support them. In this case, on the contrary, inferences are weak because of the approach chosen by the authors,  even though the type of data, and even the sample size, would support the claims made, if only data was treated differently. I think that, considering the manuscript in its current form, claims and conclusions must be softened.

Author Response

This work evaluates a potential cognitive index of the RCPM in a sample of healthy children. The findings describe different improvement trajectories according to the test's specific Factors and age range and indicate that children with higher standard scores are the ones benefiting the most from prompting, in line with previous evidence.

The manuscript is clear in most of its parts.

Authors’ reply: we thank the reviewer for this positive comment.

However, the way in which some claims were made throughout the manuscript is not appropriate given that this work followed a qualitative approach. I’ll elaborate this issue further in the comments.

Introduction

Line 30: the expression “discrimination at lower levels” is not clear to me. Can the authors rephrase this sentence?

Authors’ reply: thanks for this suggestion. The sentence is no longer present in the new revision of the manuscript.

Line 51: this sentence is missing the related references.

Authors’ reply: The related references were added following the suggestion (line 90).

Line 70: the authors hypothesized that the interview would significantly increase the performance. However, such hypotheses cannot be tested with the approach used here, but applying statistical analyses. It would be more appropriate to claim that the aim of the study was to provide an overview of the performance on the RCPM by means of descriptive statistics.

Authors' Response: Following the reviewer's helpful suggestion, we stated that the aim of the study was to provide an overview of RCPM performance by highlighting the descriptive statistics used.

Moreover, I suggest adding a part on dynamic testing.

Authors' Response: thanks to the reviewer's suggestion we have integrated a part on dynamic testing (lines 103-120)

Materials and Methods

Participants

Line 81: How can the number of subjects be counterbalanced? Does it mean that the authors kept recruiting until a certain number per age-group was reached?

Authors' Response: We mean that we have recruited until we reached a balanced number of subjects per age- group. We have clarified this point in the manuscript by following the suggestion (lines 169-170).

Procedures

Line 97: the terms Factor III and I appear here without an introduction or any explanation on what they are. I suggest adding in the method a brief description of the test, the factors and to which cognitive abilities and strategies they have been associated with in previous research.

Authors' Response: we apologize for this omission which in fact did not allow the understanding of the text. We have added a brief description of the test and an explanation of what factors are following previous factorial studies in the Methods session, as suggested (lines 182-196). This suggestion is in line with that of the third reviewer who had asked to anticipate the explanation of the factors in the introduction and we also followed this advice.

Line 97-115: I have two comments regarding the interview. First, I think it may be helpful to provide a graphic description of the procedure, for example using a flow chart, which I believe would make it quicker and easier to understand how it was conducted. But this a matter of personal preference, so if the authors prefer a verbal description, I understand their choice. Second, provide a rationale of why the structure of the interview for Factor II was different to the one of the interview for Factor I and III, for which children were not asked to explain the reason behind their response, but they were only given time to rethink their initial choice. Do the authors think that this difference may have influenced the results? If so, how?

Authors' Response: Thanks for this helpful suggestion. We have added a flowchart to also provide a graphic description of the interview (figure 1). Regarding the second comment. Considering that Factors III and I require a perceptual-based response (Factor III requires completion on the basis of similarity and identity, Factor I requires completion based on closure and directionality) and Factor II requires a high abstract reasoning process, we thought about this "very small" difference in the interview which, however, we do not think can affect the type of response, only it is more "appropriate" for the type of processing required by those items.

Line 121-126:  Please add the maximum score for each Factor to simplify the understanding of Table 1.

 Authors' Response: Thanks, we added it.

Results

Table 1: please reorder the Factors in the table.

Authors' Response: We are sorry not to be able to change the factorial order as it was identified in the previous factorial studies. They are actually numbered from previous studies, and are in order of increasing difficulty: factor 3, factor 1 and factor 2.

Line 151: The claims made here are in line with the use of descriptive statistics. However, a simple t-test – for example – would go beyond the “mere” description of higher score in the ZPD as compared to the standard condition, providing a more solid indication of whether an actual improvement has occurred or not.

Authors' Response: we have specified that the purpose and the statistical analyses are descriptive. We had chosen to present the data in terms of performance differences, because this allows readers to better observe the trend of performance differences in the different age-groups, which in any case are highly significant. As suggested, we have added the t-test data.

Discussion

Line 212-213: “performance significantly improved after interview” this claim can’t be made based on descriptive statistics. The same goes for similar claims on differences among age-groups.

Authors' Response: We believe we have "lightened" the tone of those claims, which in any case had been referred to our sample and did not want to be generalized to the entire population since ours is a study based on descriptive statistics.

Line 227-230: this part looks more of a conclusion, in my opinion, thus I suggest moving it towards the end of the discussion.

Authors' Response: that sentence has been eliminated because in the final part of the discussion there was already a sentence that stated the same.

Line 273-276: this sentence repeats what has already been said above.

Authors' Response: That sentence is no longer present in the revised version of the manuscript

Line 306-308: in line with my comment on the t-test, here a simple correlation would verify if, indeed, subjects with the highest improvement are the ones with the highest standard score, although Table 5 provides a similar indication.

Authors' Response: following the suggestion of the reviewer we added the correlation to further strengthen what table 5 said (now table 6 and correlation table 7)

Line 325: Do the authors mean “less intensive” or “less demanding” with the term lower?

 Authors' Response: yes, we apologize. We modified with less demanding (line 621)

Conclusion

Line 342: the authors refer to prompts, suggestions and cuing. I think it would be helpful to add a part, maybe in the introduction, on these procedures, on how they are used in neuropsychological testing and how they may differently influence the ZPD.  

Authors' Response: following the suggestion of the reviewer we added this part (lines 107-112)

Line 356: I’d add as a limitation the fact that only descriptive statistics were computed.

Authors' Response: we have specified better that the analysis used is a descriptive statistic which is one of the ways to analyze data referring to a sample, we do not feel like considering it a limitation.

Overall comment

This work has the potential to be of interest not only for the clinical audience but also for researchers in the field of cognitive sciences. However, the meaningfulness of the findings is strongly limited by how the data were analyzed. Sometimes researchers try to make inferences from data or analyses that cannot support them. In this case, on the contrary, inferences are weak because of the approach chosen by the authors, even though the type of data, and even the sample size, would support the claims made, if only data was treated differently. I think that, considering the manuscript in its current form, claims and conclusions must be softened.

Authors' Response: By following all the suggestions we have considerably lightened our claims and totally avoided the inferences that our type of study does not allow us. We thank the reviewer for any suggestions that we believe have helped improve the quality of the manuscript.

Reviewer 3 Report

Article Review

The article, Current and potential cognitive development in healthy children: a new approach to Raven Coloured Progressive Matrices” presents an analysis of a new way of administering the RCPM intelligence test.  Using Vygotsky’s concept of the Zone of Proximal Development, the authors suggest an approach to looking at intelligence not just as the child’s cognitive level, but also their potential cognitive level.  Examining a large sample of children between 6- and 10-years-old, the researcher found performance on the RCPM improved overall for children after a semi-structured interview that highlights potential errors the children made.  Additionally, the improvements differed by type of question and by age of the child.  The authors conclude that this process gives better insight into the child’s cognitive abilities that can maximize their developmental potential.

Overall Review

Overall, the article was well-written and clearly presented the results of a well-conducted study.  The reviewer suggests only minor revisions are needed to warrant publication.  Only XX major issues need to be addressed.

Introduction and Discussion

Comparatively, the discussion is much larger than the introduction and introduces too much new material.  Much of that new material would be better served in the introduction section.  For example, explanations of the three factors of the RCPM, how they are different, and why its important to look at differences between them should be in the introduction.  This will allow the reader a better understand of what is happening in the methods and why.

Methods

The paper does a relatively good job at describing the methodology of the study.  However, there is one part that was confusing and could be written in a clearer way.  In the statistical analysis subsection, the description of “the sum modalities for frequencies” is very confusing.  I admittedly am unclear what is being calculated here.  This then makes the analysis of these variables difficult to understand as well.  It would help to explain this in a different way.

Limitations

The description of limitations is good.  A suggestion is to include these before the conclusion section.  Otherwise, the reader ends the entire paper thinking about how the study does not measure up.  It would be a stronger paper if these came earlier and then to end on a clear statement about what we now know because of this study.

Author Response

The article, Current and potential cognitive development in healthy children: a new approach to Raven Coloured Progressive Matrices” presents an analysis of a new way of administering the RCPM intelligence test.  Using Vygotsky’s concept of the Zone of Proximal Development, the authors suggest an approach to looking at intelligence not just as the child’s cognitive level, but also their potential cognitive level.  Examining a large sample of children between 6- and 10-years-old, the researcher found performance on the RCPM improved overall for children after a semi-structured interview that highlights potential errors the children made.  Additionally, the improvements differed by type of question and by age of the child.  The authors conclude that this process gives better insight into the child’s cognitive abilities that can maximize their developmental potential.

Overall Review

Overall, the article was well-written and clearly presented the results of a well-conducted study.  The reviewer suggests only minor revisions are needed to warrant publication.  Only XX major issues need to be addressed.

 Authors’ reply: we thank the reviewer for this positive comment.

Introduction and Discussion

Comparatively, the discussion is much larger than the introduction and introduces too much new material.  Much of that new material would be better served in the introduction section.  For example, explanations of the three factors of the RCPM, how they are different, and why its important to look at differences between them should be in the introduction.  This will allow the reader a better understand of what is happening in the methods and why.

Authors’ reply: thanks for the useful suggestions. We agree with the reviewer that the explanation of the 3 factors of the RCPM should be included in the introduction and we have moved them from the discussion to the introduction. Furthermore, this observation was also made by the reviewer 2.

We have also streamlined the discussion and brought much of that "new material" that in the previous version of the manuscript was under discussion in the introductory part. We feel this balanced the two sessions very much

Methods

The paper does a relatively good job at describing the methodology of the study.  However, there is one part that was confusing and could be written in a clearer way.  In the statistical analysis subsection, the description of “the sum modalities for frequencies” is very confusing.  I admittedly am unclear what is being calculated here.  This then makes the analysis of these variables difficult to understand as well.  It would help to explain this in a different way.

Authors’ reply: we apologize if we had not written it clearly in the previous version of the manuscript, because that is simply the basis on which the means is calculated and having presented it was an immediate way to verify the differences between the two conditions of administration of the RCPM. We have written more clearly in this version of the manuscript and hopefully it will be easier for the reader

Limitations

The description of limitations is good.  A suggestion is to include these before the conclusion section.  Otherwise, the reader ends the entire paper thinking about how the study does not measure up.  It would be a stronger paper if these came earlier and then to end on a clear statement about what we now know because of this study.

Authors’ reply: we agree with the reviewer and thank him/her for this helpful suggestion. We have anticipated the limitations at the end of the discussion.

Round 2

Reviewer 2 Report

I thank the authors for having considered my comments and suggestions in revising the manuscript.

I still have a few minor requests.

Page 3, line 109-111: The sentence is not clear to me. Does “a surface association similarity” mean “association of similarities between target and entries”? What does the term “surface” mean here?

Page 4, line 172-173: I suggest adding the number of female/male participants per age-group (maybe in Table 1 or where the authors think it's more appropriate).

Figure 1: I suggest moving the box (“Before you told me...”) below “another incorrect answer”/“correct answer” and not next to them, to enhance clarity.

Page 9, line 327-329: I think this sentence needs to be rephrased. I would write something like “... shows the results of the t-tests on performance at RCPM ...”

Page 10, line 350: Please add that results are reported also for the total group, i.e., for the age-group including both below/above the mean performers. The same goes for Table 7.

Author Response

I thank the authors for having considered my comments and suggestions in revising the manuscript.

I still have a few minor requests.

Page 3, line 109-111: The sentence is not clear to me. Does “a surface association similarity” mean “association of similarities between target and entries”? What does the term “surface” mean here?

 Authors’ response: we meant just that. We have eliminated the term "surface" (line 88)

Page 4, line 172-173: I suggest adding the number of female/male participants per age-group (maybe in Table 1 or where the authors think it's more appropriate).

Authors’ response: Thanks for the suggestion we have added in table 1 the number of females/males of each age group (table 1)

Figure 1: I suggest moving the box (“Before you told me...”) below “another incorrect answer”/“correct answer” and not next to them, to enhance clarity.

 Authors’ response: Thanks for the suggestion we have moved below “another incorrect answer” / “correct answer” for clarity (figure 1)

Page 9, line 327-329: I think this sentence needs to be rephrased. I would write something like “... shows the results of the t-tests on performance at RCPM ...”

Authors’ response: Thanks, we have rephrased it following the suggestion and we think it is clearer now (line 232)

Page 10, line 350: Please add that results are reported also for the total group, i.e., for the age-group including both below/above the mean performers. The same goes for Table 7.

Authors’ response: we apologize for this oversight, we have added both in table 6 (lines 239-240, 247-248) and in table 7 (lines 251-251, 257)